# Theoretical and Experimental Study on Carbodiimide Formation

**DOI:** 10.3390/ijms25147991

**Published:** 2024-07-22

**Authors:** Marcell Dániel Csécsi, Virág Kondor, Edina Reizer, Renáta Zsanett Boros, Péter Tóth, László Farkas, Béla Fiser, Zoltán Mucsi, Miklós Nagy, Béla Viskolcz

**Affiliations:** 1Institute of Chemistry, University of Miskolc, H-3515 Miskolc, Hungary; marcell.daniel.csecsi@uni-miskolc.hu (M.D.C.); viragkondor@gmail.com (V.K.); edina.reizer@uni-miskolc.hu (E.R.); bela.fiser@uni-miskolc.hu (B.F.); zoltan.mucsi@uni-miskolc.hu (Z.M.); miklos.nagy@uni-miskolc.hu (M.N.); 2Higher Education and Industrial Cooperation Centre, University of Miskolc, H-3515 Miskolc, Hungary; 3BorsodChem Ltd., Bolyai tér 1, H-3700 Kazincbarcika, Hungary; renata.boros@borsodchem.eu (R.Z.B.); peter.toth45@borsodchem.eu (P.T.); laszlo.farkas@borsodchem.eu (L.F.)

**Keywords:** carbodiimide, isocyanate, organocatalysis, DFT, gas volumetry, reaction kinetics, activation energy

## Abstract

Carbodiimides are important crosslinkers in organic synthesis and are used in the isocyanate industry as modifier additives. Therefore, the understanding of their formation is of high importance. In this work, we present a theoretical B3LYP/6-31G(d) and SMD solvent model and experimental investigation of the formation of diphenylcarbodiimide (CDI) from phenyl isocyanate using a phosphorus-based catalyst (MPPO) in ortho-dichlorobenzene (ODCB) solvent. Kinetic experiments were based on the volumetric quantitation of CO_2_ evolved, at different temperatures between 40 and 80 °C. Based on DFT calculations, we managed to construct a more detailed reaction mechanism compared to previous studies which is supported by experimental results. DFT calculations revealed that the mechanism is composed of two main parts, and the rate determining step of the first part, controlling the CO_2_ formation, is the first transition state with a 52.9 kJ mol^−1^ enthalpy barrier. The experimental activation energy was obtained from the Arrhenius plot (ln *k* vs. 1/*T*) using the observed second-order kinetics, and the obtained 55.8 ± 2.1 kJ mol^−1^ was in excellent agreement with the computational one, validating the complete mechanism, giving a better understanding of carbodiimide production from isocyanates.

## 1. Introduction

Carbodiimides (CDIs) are versatile compounds with an R_1_–N=C=N–R_2_ chemical structure (R_1_ and R_2_ can be both aliphatic and aromatic groups), used in different industrial processes [1]. Among them, the polymer industry is an essential one since a significant amount of carbodiimide usage is required in order to improve the mechanical properties and thermal stability of the produced polymers (polyurethanes, polyesters, polyamides, etc. [1]). Regarding this, it is important to highlight the polyurethane formation process, where 4,4′-methylenediphenyl diisocyanate (4,4′-MDI) is modified to catalytically generate carbodiimide and uretonimine in a one-pot approach (the latter is produced from CDIs when isocyanate is in excess) [2,3]. This novel CDI intermediate leads to a decreased melting point of MDI (from 37 to 20 °C). Carbodiimides and polycarbodiimides are also used as fiber reinforcements, crosslinking and stabilizing agents in polymers [4,5], and furthermore as adhesives [6], even in photography as gelatin hardeners [7] and in the cosmetic industry to improve the water resistance of products like clothes and hair dyes [1,8,9]. In addition to all these, well-known carbodiimides (e.g., *N*,*N*-dicyclohexylcarbodiimide (DCC); *N*-ethyl-*N*-(3-dimethylamino) propyl carbodiimide (EDC)) are involved in the pharmaceutical industry as coupling agents for peptide and protein synthesis [10,11]. They are also present in agricultural chemicals and pharmaceutical intermediates; thus, carbodiimides affect many industrial areas and are present in numerous significant industrial methods.

Besides these applications, the interest of the scientific community in CDIs research dates back to 1874, when W. Weith, a German chemist, discovered carbodiimides by dehydrating thioureas [12]. The first review of CDIs published 70 years ago was followed by numerous books, reviews, and patents resulting in a more and more detailed and deeper understanding of the subject [13] (Figure 1).

The chemical reactions for carbodiimide formation can be distinguished based on the types of initial molecules, like thioureas, ureas, isocyanates, isothiocyanates, tetrazoles, or via other miscellaneous reactions, or, on the other hand, based on the catalysts used in the process [1,8,10,16]. Different types of catalysts are known to be used to convert isocyanate to carbodiimide, including organometallic- (arsenic, antimony, and some based on transition metals) and non-metallic phosphorous-, nitrogen-, and sulfur-based catalysts [17,20]. Phospholenes, phospholanes, their oxides and sulfides, and simple phosphine oxides (e.g., tributyl-, triphenyl-) also exhibited this catalytic effect [14,19]. In addition, it was observed that any pentavalent phosphorus compound which has a covalent bond between phosphorus and oxygen atoms catalyzes the conversion reaction, and the catalytic activity is dependent on the polarity of this phosphoryl bond and the steric hindrance around the phosphorus atom [18]. In 1962, Campbell and Monagle at Du Pont discovered that cyclic phosphine oxides are more potent catalysts for the conversion of isocyanates into carbodiimides [1]. The most effective catalyst is 3-methyl-1-ethyl-3-phospholene-1-oxide [15,19]; however, its 1-phenyl analogue (3-methyl-1-phenyl-2-phospholene-l-oxide, MPPO) is less active but more easily accessible. Considering the solvent effect, benzene proved to be sufficient among the aromatic solvents since small molecular structural differences do not influence kinetics, except for solvents with a high dielectric constant [15,19]. The carbodiimide formation process not only produces the main product CDIs but also carbon dioxide as a significant byproduct; nevertheless, other side reactions produce different species. These side products are the dimer and trimer of isocyanate [26], CDIs dimer, ref. [1] or the more significant uretonimine (or uretidirnone) [3,27], which is formed mainly in the later phase of the reaction. This prior knowledge provided by the literature allowed us to design the optimal experimental conditions.

In our previous works, a combined theoretical and experimental study of the catalytic urethane bond formation is presented [25,28]. The computational results showed that tertiary amines with low steric hindrance around the nitrogen atoms proved to be optimal catalysts for urethane bond formation. In addition, a fairly good correlation was obtained between the calculated and kinetically measured activation energy values, which confirms the reliability of both the developed theoretical and experimental methods [25,28]. The investigation of the alicyclic derivatives of CDIs and the protonation study of some similar heteroallenes are also presented in theoretical studies using the MP2 level of theory [24,29]. Although the bond formation is mostly a catalytic process, detailed recipes are known to produce diphenylcarbodiimide without a catalyst, therefore involving a catalyst in dry benzene solvent and under nitrogen stream too [15]. The reaction kinetic background of CDIs formation also originates from Monagle and his co-workers, who observed not only the substituent effect of isocyanates and the solvent effect but also the kinetic dependence on the catalyst and isocyanate concentration [16]. The rate determining step is the formation of phosphine imide, which is driven by the exiting CO_2_ molecule. The activation energy of carbodiimide formation using different solvents was found to be 12.40 ± 1.45 kcal mol^−1^ (51.9 ± 6.1 kJ mol^−1^) [16]. Nevertheless, besides calculating the structural energies of the isomers and conducting separate kinetic studies, detailed computational mechanistic analysis and comparison with the reaction kinetics of the CDIs bond formation has not been mentioned previously.

Although bifunctional isocyanates are more significant and used often in industry to prepare CDIs, here, we examined phenyl isocyanate (referred to as NCO in the following), which is a simpler and better candidate to study the formation of diphenylcarbodiimide (referred to as CDI in the following), according to Figure 2.

Several preliminary experiments were carried out to establish the appropriate experimental parameters. At first, the solvent effect was investigated, where experiments were performed in orthodichlorobenzene (ODCB), toluene, and acetonitrile and in the absence of solvent. Among them, the NCO conversion in ODCB proved to be more beneficial since it models the conditions used in the industrial production of MDI [30]. We tested several potent catalysts. Relatively low conversion values were observed for titanium isopropoxide [Ti(OCH(CH_3_)_2_)_4_] with the appearance of an unexpected reddish color (may be TiO_2_ sol [31]) with several byproducts [32]. The more efficient phosphorus-based catalyst, MPPO, was favored, as seen in Figure 2 above.

The purpose of our work is to experimentally and theoretically study the carbodiimide bond formation through a catalytic reaction, having two monoisocyanate reactants developing CO_2_ as a byproduct (Figure 2).

## 2. Results and Discussion

### 2.1. Theoretical Investigation and General Concerns

Although reaction mechanisms for the formation of diphenylcarbodiimide (CDI) from phenyl isocyanate (NCO) have already been proposed in the literature [9,19], they are flawed and fail to explain the whole process in detail. Here, we present a more detailed catalyzed mechanism, calculated by the B3LYP/6-31G(d)[SMD(ODCB)] level of theory, which can be split into two subprocesses and is summarized in Figure 1a,b, Appendix A and Table 1. In the first subprocess (Figure 1a), the **NCO** attacks the **MPPO** catalyst, releasing CO_2_ and forming phosphine imide (**IM**). Figure 1b shows the second subprocess of an additional **NCO** attack and the final formation of the product **CDI** as well the catalyst recovery of the **MPPO** catalyst. The structures of the possible intermediates (**IMs**), adducts (**ADs**), complexes (**RC**, **PC**), and transition states (**TSs**) were calculated.

The first elementary step of the mechanism involves the formation of reactant complex (**RC**) from an **NCO** and an **MPPO** catalyst molecule. Subsequently, the electrophilic carbon atom of the isocyanate interacts with the **MPPO**’s oxygen atom forming a bond via a transition state (**TS1**). The initial C–O distance is more than 3 Å, which is reduced in the **TS1** transition state to 1.863 Å, and finally, the C–O bond is formed in **AD1** with a 1.504 Å bond length. The next step involves an internal bond rearrangement via a turnstile pseudorotation [33,34] around the P atom (**AD1** → **TS1** → **AD2**) in the four-membered P–O–C–N structure, where the axial N atom occupies the equatorial position (Figure 2). In **AD1**, *d*(P–O_eq_) = 1.608 Å and *d*(P–N_ax_) = 2.633 Å, while in **AD2**, the new P–N_eq_ bond is formed with the sterically hindered nitrogen of *d*(P–N_eq_) = 1.688 Å, while the P–O_ax_ bond is stretched further to the distance of *d*(P–O_ax_) = 2.351 Å. As a consequence of this rearrangement, the carboxylate group is now easily detached from this axial position as CO_2_ in **TS3** resulting in a loosely bonded molecular complex (**IM1**). This finally leads to the stable phosphine imide intermediate (**IM**). The CO_2_ evolution was measured during the kinetic experiments.

In the second subprocess, the **IM** forms a new complex with an additional isocyanate molecule, yielding **IM2**. Unlike the first subprocess in Figure 1a, in the second subprocess, the nucleophilic atom is the nitrogen of the **IM** structure. Although a similar P–O–C–N structure is formed compared to **AD1**, here, the C=O moiety of the second entering **NCO** molecule plays a crucial role in establishing the P–O–C–N ring structure, in contrast to the previously showed N=C. Through **TS4,** a new C–N bond is formed with *d*(C–N_eq_) = 1.450 Å in **AD3** (Figure 1b). It is followed by an analogous pseudorotation via **TS5**, where the longer P–O_ax_ bond is shortened to P–O_eq_ = 1.706 Å in **AD4**; meanwhile, the P–N_eq_ is stretched to P–N_ax_ = 1.999 Å. It should be differentiated that although the transition movement of the large **MPPO** group is the same in both **TS2** and **TS5**, the bonds wander from P–O to P–N in **TS2** while from P–N to P–O in **TS5**. Nevertheless, the last step is the formation of the **PC**, product complex, via **TS6**, where the C–O bond is broken, while P=O (1.514 Å) and C=N (1.225 Å) double bonds are formed simultaneously, where the **MPPO** catalyst recovers and the expected **CDI** product is formed.

To understand more deeply the electronic structures of the ring intermediates (AD1–4) and TSs, the concept of Nucleus-Independent Chemical Shift (NICS, *δ*_0_) was also calculated by the standard method. Aromaticity often brings extra stability for unsaturated heterocycles [35]. Phosphorous unsaturated four-membered heterocycles exhibited interesting aromatic or antiaromatic characteristics, depending on their pattern, and several examples were studied [35,36,37]. In all cases, the different P–O–C–N ring structures had a slightly aromatic characteristic (between −3.8 and −6.5 ppm, see Table 1); the largest values can be attributed mainly to **AD1** and **TS5**. This aromatization stabilizes the intermediates and facilitates their interconversion. However, it should be mentioned that in these narrow, four-membered rings, the σ shielding of the core electrons of the atoms is significant, causing larger and overestimated values. Moreover, the orientation of phenyl groups may significantly affect the relative energies. For instance, initially, **MPPO**’s phenyl group is oriented to the plane of the P=O double bond since it is the consequence of the set-up hydrogen bond between the hydrogen of phenyl and oxygen of P=O. This position remains the same up to the **AD1** structure, where the **TS2** transition changes it to the plane of P–N in **AD2**. This also lasts until it rotates back in **TS5** to the bond plane of P=O, creating **MPPO** again. The ring strain energy is significant for the four-membered ring, compensated by the stabilization energy of the aromatic system [38].

### 2.2. Energetics

Thermodynamic calculations were carried out to determine the relative energy values compared to the starting point (Table 1). Solvent phase calculations in ODCB included three specific thermodynamic values: enthalpy (Δ*H*), Gibbs free energy (Δ*G*), and the sum of electronic and zero-point-corrected energy (Δ*E*_0_; see Table 1). In the following, the more appropriate and exact enthalpies are used in the discussion and conclusion. Enthalpies and Gibbs energies were calculated at room temperature (298.15 K) and atmospheric pressure, which was previously found to be reliable for modeling similar reactions [25,39]. Furthermore, the potential energy surface (PES) diagrams with enthalpies (Figure 3), Gibbs free energies, and the sum of electronic and zero-point-corrected energies (Appendix A) are also presented.

The starting point is the reactant reagents complex of **NCO** + **MPPO**, which is defined as 0.0 kJ mol^−1^ as a reference. The process begins with the first subprocess, with the formation of the **RC**, which is slightly higher in enthalpy (3.2 kJ mol^−1^; Table 1 and Figure 3), which is attributed to the structural similarities. The first transition state (**TS1**) is formed with a moderately high relative enthalpy (43.4 kJ mol^−1^), followed by the formation of a stable **AD1** complex in the next step. Subsequently, via a moderate **TS2** (48.3 kJ mol^−1^), the **AD2** complex lowers the enthalpy level to −28.2 kJ mol^−1^. Through the second highest enthalpy **TS3** (52.9 kJ mol^−1^), the **IM1** complex is formed when CO_2_ is eliminated from the structure. Interestingly, from an enthalpy point of view, a less stable **IM** is formed compared to **IM1** and **IM2**; however, the leaving of the CO_2_ provides a high entropy release and makes the formation of the IM irreversible. To keep appropriate relative energies, the presented **IM** enthalpy is given by the sum of calculated **IMs** and the leaving CO_2_ energy, with the difference in initial reactants. Similarly, **IM2** is formed by subtracting the entering **NCO**’s energy, which is already presented in the complex. **IM2** is pursued by the highest energy barrier toward **TS4** (Δ*H* = 79.1 kJ mol^−1^). In the literature, this high energy barrier was explained by the low reactivity of the phosphine imide (**IM**), resulting in a slow reaction with the **NCO** [1], which agrees well with our calculations. In the second subprocess, analogously to the previous steps, the enthalpy barrier is followed by a deep valley of **AD3**, with a relative enthalpy of 12.5 kJ mol^−1^, leading to the **AD4** complex (Figure 3) via **TS4**. In the next step, **TS5** (52.7 kJ mol^−1^) is developed towards the final and most stable complex of **PC** (−19.0 kJ mol^−1^; Table 1). Finally, the **P**, product, includes the relative energy of **CDI** + **MPPO**. Nonetheless, the sum energy of products remains negative, which declares an exothermic process of Δ_r_*H* = −15.1 kJ mol^−1^ reaction enthalpy, and the negative Gibbs free energy (Δ_r_*G* = −6.7 kJ mol^−1^) also demonstrates a thermodynamically spontaneous, one-way catalytic reaction (Table 1, Appendix A).

The theoretical results prompted us to investigate the reaction under experimental conditions and to determine the activation energy from reaction kinetics in order to compare with the theoretically derived one.

### 2.3. Kinetic Results

Reactions were carried out at five different temperatures between 313 and 353 K, with the measurement time range varying between 12,600 s and 5400 s, respectively. It should be noted here that the calculated **NCO** conversion is directly related to carbodiimide (**CDI**) formation, and no formation of side products was taken into consideration. The **NCO** conversion versus time data (Figure 4) reveal that in each case the plots run into saturation, i.e., equilibrium is reached. It is evident from Figure 4 that the reaction has a significant temperature dependence. At 353 K, almost 100% conversion is reached within 4800 s, while at 313 K, the conversion is approximately 55% even after 11,000 s. It can also be seen that **NCO** conversion is directly proportional to temperature: the higher the temperature, the higher the conversion (observed independently at all time values).

The experiment reveals that the position of the equilibrium is strongly dependent upon the temperature. The lower equilibrium constants (lower conversions) suggest a reaction step with a relatively high activation energy value. Based on Figure 4, it can also be concluded that the optimal reaction temperature is 80 °C (353 K) for 1.5 h (5400 s) since the reaction goes to completion in the ODCB solvent under these conditions.

Second-order kinetics was applied to describe the process since the reaction fits the 2A → product + byproduct model, i.e., it proceeds through the reaction of two **NCO** molecules. The differential Equation (1) and integrated rate Equation (2) are shown below, where [**NCO**] is the actual isocyanate concentration at time *t* in mol dm^−3^, *t* is the time in seconds, *k* is the rate constant in mol^−1^ dm^3^ s^−1^, and [**NCO**]_0_ is the initial concentration of isocyanate in mol dm^−3^.
(1)$$-\frac{1}{2}\cdot \frac{d\left[NCO\right]}{dt}=k{\left[NCO\right]}^{2}$$
(2)$$\frac{1}{2\left[NCO\right]}=\frac{1}{2[NCO{]}_{0}}+k\cdot t$$

Kinetic diagrams (Figure 5a) were constructed based on Equation (2) using the method of initial rates. At higher conversions, the rate equation may not be second-order due to the formation of side products, as was described in the Introduction. An almost perfect linear fit (*R*^2^ = 0.99) of the experimental points can be observed at each temperature, further supporting the second-order kinetics of the reaction. The rate constants (*k*) were obtained as the slope of the fitted linear lines and are collected in Table 2. As seen in Figure 5a, the lines intercept the *y* axis at almost the same value around 0.50 since they show similar initial concentrations. The slope of the fitted lines increases with the increasing temperature, and as the data of Table 2 indicate, the rate of the reaction at 80 °C is tenfold that at 40 °C (*k*_(313K)_ = 1.67 × 10^−4^ M^−1^s^−1^ vs. *k*_(353K)_ = 17.9 × 10^−4^ M^−1^s^−1^). Detailed nonlinear curve fits were also carried out (Appendix A), where the obtained deviation of rate constants is at most only 5% compared to the linearized regressions (Table 2, Appendix A). In order to ensure the reproducibility of measurements, the reactions were repeated at least two times at each temperature, and the rate constants showed a good match.

The activation energy (*E*_a_) can be calculated using the linearized Arrhenius equation, when plotting ln *k* as a function of 1/*T* (Figure 5b). The activation energy was found to be *E*_a_ = 55.8 ± 2.1 kJ mol^−1^ in the observed 40–80 °C temperature interval, a value that agrees excellently with previously published data for a similar reaction (51.9 ± 6.1 kJ mol^−1^ [16]).

## 3. Materials and Methods

### 3.1. Theoretical and Computational Methods

Theoretical calculations were performed by using the Gaussian 16 program package [41]. The structures for the reaction mechanism, including reactants, products, intermediates, and transition states, were optimized by density functional theory (DFT) methods. B3LYP hybrid functional [42] with a 6-31G(d) basis set was used in combination with the SMD implicit solvent model [43] for ODCB (*ɛ*_r_ = 9.9949). At first, gas phase calculations were studied; then, a higher level of theory was also applied to improve the accuracy of the calculations. However, not all structures were able to be determined; thus, B3LYP/6-31G(d) was favored. When normal transition state calculations did not result in the right structure, the QST3 method was used to find the appropriate critical saddle point. The found-out transition states were validated by performing intrinsic reaction coordinate (IRC) calculations. Furthermore, standard Nucleus-Independent Chemical Shift (NICS) calculations [38] were carried out at the same level of theory for the relevant structures to describe the aromaticity of the determining P–O–C–N ring. The 2D potential energy surfaces (PESs) and thermodynamic properties of the reaction mechanism were also analyzed.

### 3.2. Materials and Apparatus

Kinetic measurements were performed using phenyl isocyanate (NCO, 99%, Acros Organics, Geel, Belgium) as a reactant, ortho-dichlorobenzene (ODCB, from industrial source; see purity in Appendix A) as a solvent, and 3-methyl-1-phenyl-2-phospholene-l-oxide (MPPO, INTATRADE Chemicals, Muldestausee, Germany; see recipe in [44]) as a catalyst. The reaction chamber was flushed with nitrogen gas (99.996%, Linde, Repcelak, Hungary) to establish an inert atmosphere within the system. The gas burette and level vessel (Figure 6) were filled with acidified, saturated Na_2_SO_4_ solution to reduce the absorption of carbon dioxide. The solution was colored with methylene blue for better readability.

### 3.3. Experimental Methods and Calculations

To ensure low side-product ratios and accurate traceability through measurements, the reaction mixture contained 10 wt% of NCO, which proved to be enough for the investigations. The low-scale reaction mixture of merely 10.00 g was composed of 0.02 g MPPO, 8.98 g ODCB, and 1.00 g NCO. The schematic representation of the experimental apparatus (Figure 6) and the detailed procedure are presented below.

A total of 8.00 g of the premixed MPPO solution in ODCB was introduced into a glass-jacketed reactor (F). Since kinetic measurements are highly sensitive to changes in reaction temperature, a precision thermostat (A) was used to keep the reaction mixture at the predefined temperatures of 40, 50, 60, 70, and 80 °C (shown in Kelvin in diagrams). The reflux condenser (D) was set at 25 °C with water flow (B) to keep the evolved gas at a constant temperature. During stirring, the system was flushed with nitrogen, and consequently, 2.00 g of NCO/ODCB solution (it is easier to handle) was added rapidly through the inlet (E) by using a dropping funnel with a pressure equalizing tube, connected to the reactor (this way, there is no need to open the system, just turn the stopcock). Simultaneously, a three-way glass stopcock (H) was changed from outer atmosphere (I) to the gas burette (J) position, and a stopwatch was started. The reaction temperature was continuously monitored via a digital thermometer (C). To eliminate the formation of overpressure inside the system, which could negatively influence CO_2_ evolution at all of the temperatures, a compensative level flask (K) was applied, the position of which was altered continuously and precisely with a laboratory scissor jack (L). This simple, depicted apparatus let us observe the reaction accordingly and precisely.

During measurements, the volumes of CO_2_ evolved were read on the gas burette and were converted into moles ($n_{CO_{2}}$) using the ideal gas law at *p* = 101,325 Pa and *T* = 298.15 K (standard conditions). To obtain the exact isocyanate concentration [NCO] at time *t* in the vessel, current reaction volume data (*V*_actual_) are required. The actual volume of the reaction mixture was derived using Equation (3):(3)$$V_{actual}=\frac{m_{0}-{(n}_{CO_{2}}\cdot M_{CO_{2}})}{\rho },$$
where *m*_0_ is the mass of the reaction mixture at *t*_0_, ($n_{CO_{2}}\cdot M_{CO_{2}}$) is the mass of CO_2_ leaving, and *ρ* is the density of the model solution. The density of a 10 wt% phenyl isocyanate containing model solution was determined experimentally at room temperature using a digital scale and was found to be *ρ* = 1.275 g/cm^3^. It should be noted that the temperature dependence of the density was neglected regarding the reaction temperatures [16]. Consequently, the actual isocyanate concentration is given as [NCO] = $\frac{{n_{0}-n}_{NCO}}{V_{actual}}$ and the conversion as $X=\frac{n_{NCO}}{n_{0}}$, where the reacted NCO moles ($n_{NCO}$) are given by the stoichiometry of the reaction in Figure 2, that is, $n_{NCO}=2n_{CO_{2}}$, while $n_{0}$ is the initial NCO concentration.

## 4. Conclusions

In this work, we present a more detailed reaction mechanism of carbodiimide formation from isocyanates in the presence of a cyclic phosphine oxide catalyst, based on theoretical and kinetic experiments in ODCB solvent. To investigate the reaction mechanism, a computational study was carried out on the B3LYP/6-31G(d) level of theory accurate for solvent phase calculations. The reaction mechanism was found to be composed of two subprocesses, involving six steps overall: eight complexes and six transition states, where a special transition was observed too, when a P–O–C–N aromatic ring was formed and bond rearrangements occurred. Thermochemical data (including enthalpy, Gibbs free energy, and zero-point energy) and potential energy surfaces were ascertained. The enthalpy barrier between R and TS3 refers to the theoretical activation energy of the reaction, which is the rate determining step through the formation of phosphine imide and carbon dioxide evolution, with an enthalpy of Δ*H* = 52.87 kJ mol^−1^. The second half of the reaction includes another NCO addition, and the MPPO catalyst recovers. In addition, the reaction enthalpy of −15.05 kJ mol^−1^ shows an exothermic process. To confirm the theoretical (DFT) findings, a simple but reliable gas volumetric apparatus was developed to determine the kinetic parameters of the reaction. In line with the theory, second-order kinetics were observed, and using the method of initial rates, rate constants were determined. Furthermore, based on the Arrhenius formula, the activation energy was also obtained using both linear and nonlinear curve fitting, where the activation energy was found to be *E*_a_ = 55.8 ± 2.1 kJ mol^−1^ in the observed 40–80 °C temperature interval. Furthermore, we proved that this obtained activation energy falls within a very good 4 kJ mol^−1^ interval compared to previously published experimental data, as it is only the error of plotting. Nevertheless, it is more important to highlight that, as the energy was determined in two ways, the deviation between the calculated (DFT) and experimental activation energy is only 5.4%, which confirms the validity of the proposed mechanism and the practicability of the experimental apparatus. In summary, these results may help to better understand the behavior of carbodiimide formation at both a laboratory and industrial scale.

## Data Availability

All data are available from the corresponding authors upon reasonable request.

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
