# Peer review of "Theoretical and Experimental Study on Carbodiimide Formation"

_ijms, 2024, doi:10.3390/ijms25147991_

Round 1

Reviewer 1 Report

Comments and Suggestions for Authors

Comments:

This study investigates the formation of diphenylcarbodiimide (CDI) from phenyl isocyanate using a phosphorus-based catalyst in ortho-dichlorobenzene solvent. Combining theoretical (DFT calculations with B3LYP/6-31G(d) and SMD solvent model) and experimental approaches, the research unveils a detailed reaction mechanism and validates it through kinetic experiments. This comprehensive understanding enhances the synthesis of carbodiimides, crucial for advancing organic synthesis and industrial applications in modifying isocyanates.

Some modified suggestions are as follows,It is recommended that the author carefully revise it:

1.     Scheme 1a and Scheme 1b are the “Detailed reaction mechanisms”. The author gives the transition state structure. It is recommended that the author consider adding the chemical structure diagram of the transition state and adding arrows to indicate the entire reaction process. This will make it easier for readers to understand how the reaction process occurs.

2.     Generally speaking, the energy of reactant A plus the energy of reactant B should be higher than the complex energy of reactants A and B, so that A and B have better interactions and there is a tendency for A and B to bind. Why does Figure 4 show that the energy of RC is higher than that of R?

3.     What is the basis for choosing 6-31G(d) in the calculation part of the paper? Did the author refer to other basis sets? Since the author has considered solvation, it is recommended that the author consider large basis set correction and dispersion correction, etc.

Comments on the Quality of English Language

English is in general good.

Author Response

Please, see attached file.

Reviewer 2 Report

Comments and Suggestions for Authors

Theoretical and experimental study on carbodiimide formation is very interesting paper! Some improvements are required.

Page 1:

Abstract

at different temperature (at different temperatures in range between….)

The experimental activation energy was obtained from the Arrhenius plot (ln k vs 1/T) using the observed second-order kinetics and the obtained 55.8 ± 2.1 kJ mol–1. In which temperature is this Activation anergy valid?

This novel CDI intermediate leads to a decreased melting point of MDI. (what is a value of a decreased melting point?)

Page 2:

This prior knowledge provided by literature allowed us the design of optimal experimental conditions.Which experimental parameters are most important for this analysis?

Page 5:

To eliminate the formation of overpressure inside the system (at which temperature?)

Page 10:

Reaction Kinetics

Reactions were carried out at five different temperatures between 313 and 353 K (which reaction time?)

It can also be seen that NCO conversion is directly proportional to temperature: the higher the temperature, the higher the conversion (in which reaction time?)

Page 11:

The activation energy was found to be Ea = 55.8 ± 2.1 kJ mol-1 (in which temperature interval?)

Page 12:

Conclusion

Furthermore, based on the Arrhenius formula activation energy was also obtained using both linear and nonlinear curve fitting, where the activation energy was found to be Ea = 55.8 ± 2.1 kJ mol-1  (In which temperature interval?)

Author Response

Please, see attached file.

Reviewer 3 Report

Comments and Suggestions for Authors

I am positively impressed by this work and suggest its publication in the International Journal of Molecular Sciences.

The authors present an exciting combination of very practical and applied work enriched with theory. The work can be accepted in its current form.

Author Response

Comment: I am positively impressed by this work and suggest its publication in the International Journal of Molecular Sciences.

The authors present an exciting combination of very practical and applied work enriched with theory. The work can be accepted in its current form.

Response: Thank you for recommendation, we highly appreciate your thoughts about our manuscript.